# A Post-Analysis of the Introduction of the EU Directive 92/57/EEC in the Swedish Construction Industry

**Leif Berglund, Jan Johansson** 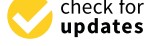**, Maria Johansson, Magnus Nygren \*, Björn Samuelson and Magnus Stenberg**

Department of Human Work Science, Luleå University of Technology, SE 971 87 Luleå, Sweden
\* Correspondence: magnus.nygren@ltu.se

**Abstract:** The EU directive 92/57/EEC focuses on ensuring that health and safety-related matters are taken into consideration during every stage of construction-related work and has been introduced into the regulations of the member countries. In 2006, Sweden was tasked by the European Commission to clarify its implementation of the directive, including which management roles and responsibilities were to come into effect during both the planning and eventual execution of construction work—changes that ultimately were introduced into the national regulations in 2009. Focusing on the accident trends in the construction industry in the years immediately following these regulatory changes, we find that the new management roles and responsibilities had no apparent effect on the accident rates. Furthermore, we argue that there is a need to broaden the analysis regarding the implementation of the EU directive 92/57/EEC to also include nation-specific changes to health and safety management and policy. These qualitative studies should also include a dedicated focus on how changes to management structures and processes may affect the prevalence of occupational diseases specifically.

**Keywords:** occupational health and safety; occupational accidents; occupational diseases; EU directive 92/57/EEC

## 1. Introduction

The matter of construction-related accidents showing a positive development over time in terms of lowered rates has received attention in the literature in recent decades. Examples of this include a study by Lopez et al. [1] showing that fatal accidents have declined in recent decades in both the Spanish and the US construction industry. Similar developments can also be seen in the Swedish construction industry [2] as well as in Australia with regard to both fatality and incidence rates [3]. A related development can also be seen in Hong Kong, where the downward trend in accident rates has plateaued in the last decade [4].

Focusing on Sweden specifically, a particular regulatory change may explain part of this long-term development, namely the introduction of new regulations for health and safety with regard to the construction industry. This change is connected to the European Union (EU) directive 92/57/EEC which member countries have had to implement, focusing on the requirements to proactively ensure the health and safety on construction sites. However, Martinéz-Aires et al. [5], focusing on the early implementation of the directive in fifteen EU countries, showed that Sweden had an increase rather than a decrease in accident rates between 1995 and 2005. Seemingly, the early implementation of the directive did not contribute to a lowering of the accident rates in the construction industry. In 2006, Sweden was criticised by the European Commission regarding its implementation of certain parts of the directive. This included a need for clarification of which roles and responsibilities were to come into effect during both the planning and eventual execution of construction work, more specifically in relation to the coordination of the different activities involved. These roles and responsibilities were ultimately introduced into the relevant regulations in 2009 and were expected to further bring about a positive development in

terms of lowered number and rates of accidents by, for example, placing greater emphasis on the importance of prioritizing health and safety already at the planning stage [6]. The matter of considering health and safety in the early stages of construction-related work has also been addressed in the literature in recent years focusing on the concept of prevention through design, e.g., [7–9]. However, when addressing the general trend in the European Union for the period following 2009 (see Figure 1), we can conclude that the changes had no apparent effect on the accident development after their implementation, i.e., from 2009 and onwards. Although focusing on a different time period, this is in line with the conclusions by Martinéz-Aires et al. [5] regarding Sweden's implementation of the original directive in the 1990s and the fact that it seemingly did not contribute to a lowering of accident rates in the construction industry.

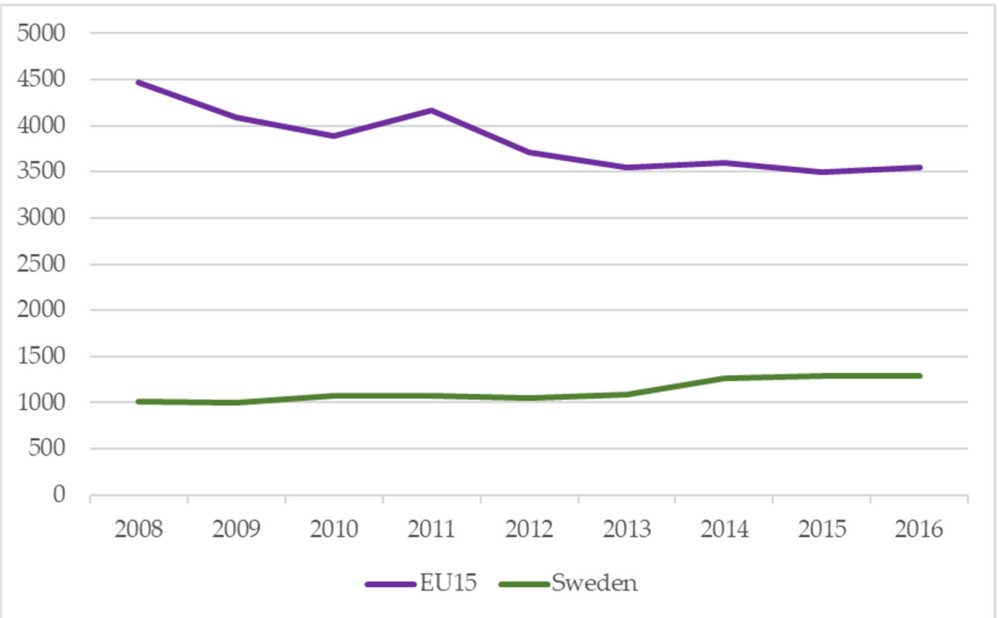

**Figure 1.** Number of serious non-fatal accidents at work in construction industry per 100,000 employees from 2008 to 2016. EU15 comprises Austria, Belgium, Denmark, Finland, France, Germany, Greece, Ireland, Italy, Luxembourg, Netherlands, Portugal, Spain, Sweden, and United Kingdom. [10], n.d.

Previous research on the implementation of the EU directive has focused on matters such as complex and diffuse formulations in the regulations themselves, which have made the directive difficult to implement in practice [11,12]. According to Almén and Larsson [6], in the seemingly only available research article highlighting the Swedish context, health and safety coordinators had varied experiences in the early 2010s regarding the implementation of the new roles and responsibilities. The authors conclude that there was a need for clarification from a regulatory standpoint regarding what these roles should ultimately entail when it came to, e.g., formal competence. Overall, the effectiveness of the changing regulations has thus been questioned in previous research.

However, an important aspect to note when analysing accident development over time is that the focus tends to be on data on an industry-wide level, i.e., with all the activity in the sector taken together as a whole [1,13–15]. This may lead to a loss of nuance regarding the development in different facets of the industry. For example, the above-mentioned changes to management roles and responsibilities may have a more significant impact on specific trades and occupational groups that are involved in large-scale projects, i.e., in situations wherein the division of roles and responsibilities may be more complex due to the involvement of multiple companies in subcontracting chains.

The purpose of this article is thus to explore more detailed accident statistics on a trade level between 2009 and 2016, and to discuss these developments in relation to the implementation of the new roles and responsibilities for health and safety management

during the same time period, as a consequence of the EU directive 92/57/EEC. The goal of the article is to further contextualise the implementation of the EU directive and the subsequent additions in the form of new management roles and responsibilities in Sweden. As noted by Weil [16], government interventions to improve the conditions in the workplace have historically proven to be a difficult task. In this article, we broaden the discussion to also include interventions with a basis in supranational regulation and policy.

In the following, the methodology is described and the trade structure of the Swedish construction industry is outlined, see Figures 2 and 3. After that, accident statistics for the twelve subtrades are presented, see Figures 4–15, followed by a comparison of similarities and differences between them. Finally, the results are discussed and recommendations for further studies are made.

## 2. Materials and Methods

The article includes all accidents that led to at least one day of absence from work between 2009 and 2016 in the twelve subtrades that comprise the trade of construction and civil engineering, as reported to the Swedish Social Insurance Agency (SSIA). After an accident has occurred, the employer of the injured individual sends a form to the agency containing information about the incident. This form is then transmitted to the Swedish Work Environment Authority (SWEA), which is the national agency that regulates occupational health and safety in Sweden. The reported accidents also include those that occurred when traveling between workplaces during work hours. The data used for this article were collected by one of the authors with the aid of representatives from the SWEA [17–24].

The data that are reported to the SSIA and the SWEA include details of the injured person, such as the extent of the injury, as well as the underlying causes of the incident in question and information regarding the employer. If an individual who is not a Swedish citizen suffers an accident while working for a foreign company, this is not included in the statistics, given that these individuals are formally insured in their home country. However, Swedish workers who are involved in an accident while working in another country are included in the data. It is furthermore important to note that the present paper explores the accidents that were reported to the SSIA, i.e., not taking into consideration whether or not a given accident was ultimately approved for injury compensation [24].

This paper focuses on the incidence rates of the above-mentioned data, i.e., number of accidents per 1000 employees, which is presented in Figures 4–15. To illustrate the development of incidence rate over time, we also calculated the linear progression for each subtrade. The statistics concerning the number of employed individuals were collected from Statistics Sweden's Register-Based Labour Market Statistics, see Figures 2 and 3. This is an annual survey based on the self-declarations of those who are self-employed as well as the employers' control data. The subtrades were grouped according to the so-called SNI2007 code, which is the formal group classification format in Swedish working life with a basis in employment structures [25].

*The Trade Structure of the Swedish Construction Industry*

In most countries, the construction industry is divided into a number of specialized trades and related occupational groups. In Sweden, the industry is structured into eight separate trades that are active in particular types of construction-related work: construction and civil engineering, sheet-metal roof covering, electrical installations, ventilation, plumbing, painting, glazing, and machine contracting [26]. Out of these trades, construction and civil engineering is dominant with 190,958 employees in 2016 or approximately 56% of the total workforce in the industry as a whole.

Construction and civil engineering, in turn, is divided into twelve subtrades that vary in terms of number of employees, although each subtrade has grown in size to various extents between 2009 and 2016. The subtrades are: development of building projects/construction of residential and non-residential buildings, construction of roads and

motorways, construction of railways and underground railways, construction of bridges and tunnels, construction of water projects/construction of other civil engineering projects n.e.c. ("not elsewhere classified"), construction of utility projects for fluids/construction of utility projects for electricity and telecommunications, demolition, site preparation/test drilling and boring, plastering, floor and wall covering, erection of other roof covering and frames, and various other specialised construction activities n.e.c. A number of subtrades, most notably development of building projects/construction of residential and non-residential buildings, are significantly larger than the others (Figure 2). The rest of the subtrades have had close to, or under, 2000 employees (Figure 3).

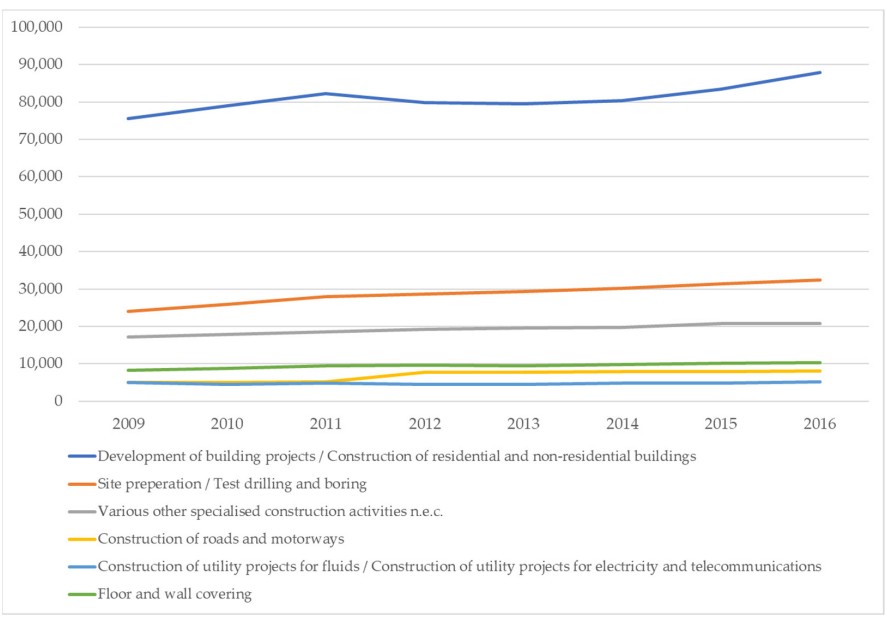

**Figure 2.** Number of employees in the six largest subtrades, 2009–2016. Based on data from [27].

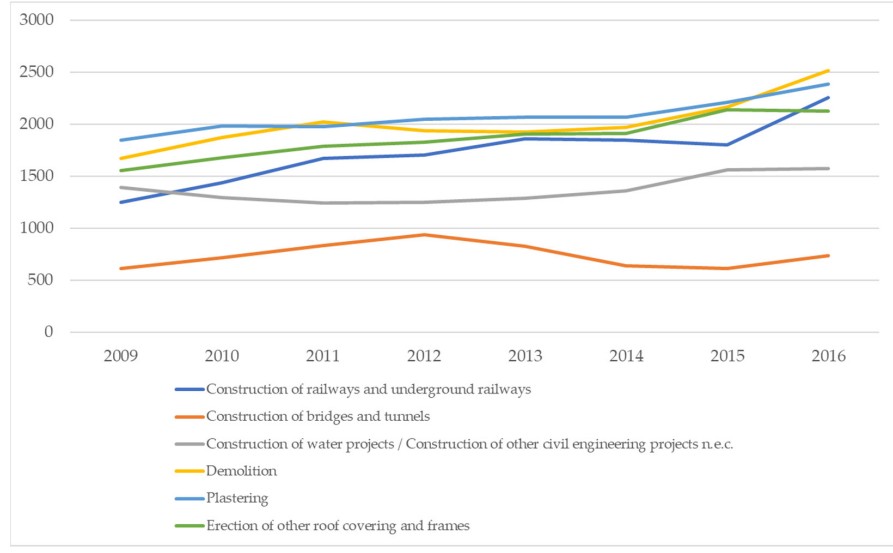

**Figure 3.** Number of employees in the six smallest subtrades, 2009–2016. Based on data from [27].

## 3. Results

In this section, the incidence rates for the twelve subtrades that comprise Construc-tion and civil engineering are shown in figures focusing on total incidences rates, as well as incidence rates for 1–14 days of absence and more than 14 days of absence, respectively.

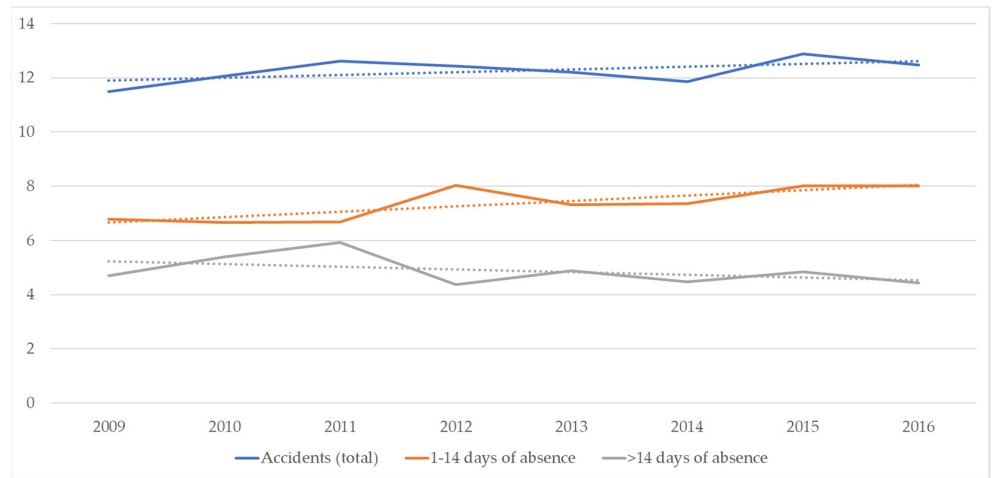

**Figure 4.** Incidence rates in development of building projects/construction of residential and non-residential buildings. Based on data from [17–24,27].

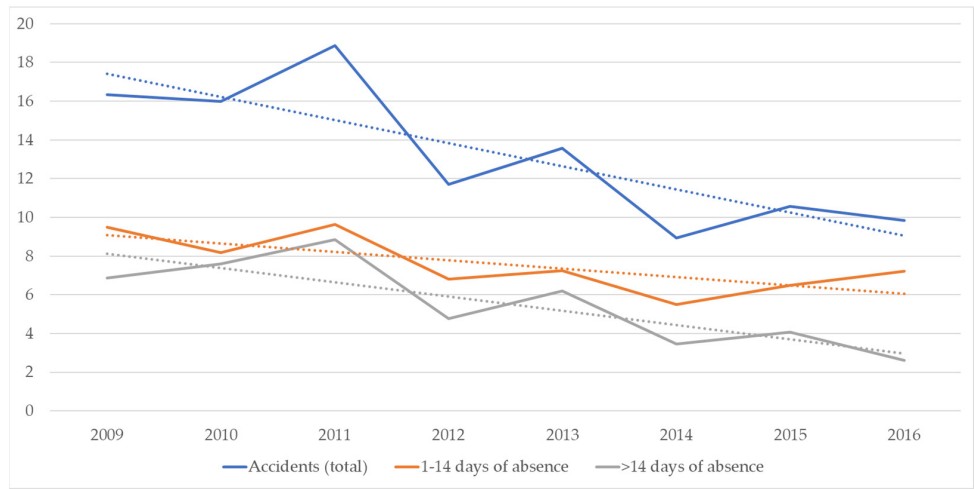

**Figure 5.** Incidence rates in construction of roads and motorways. Based on data from [17–24,27].

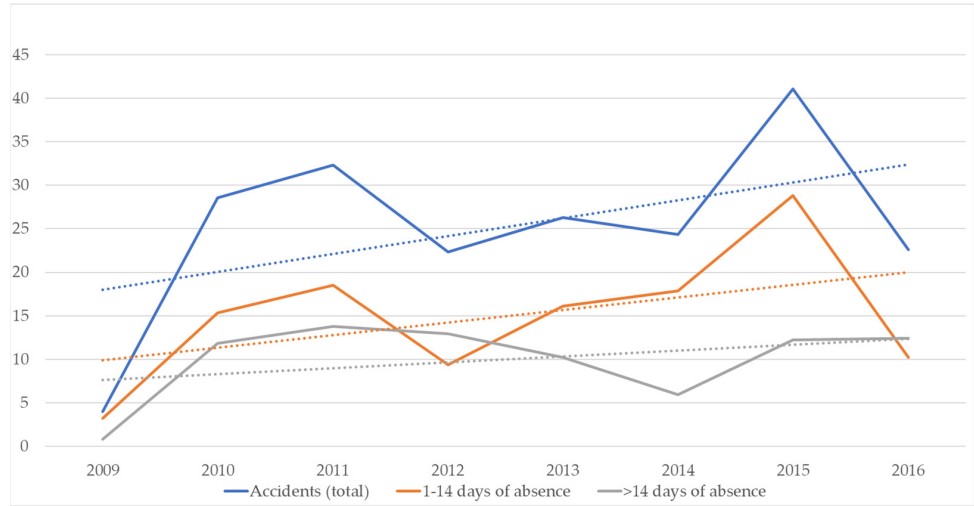

**Figure 6.** Incidence rates in construction of railways and underground railways. Based on data from [17–24,27].

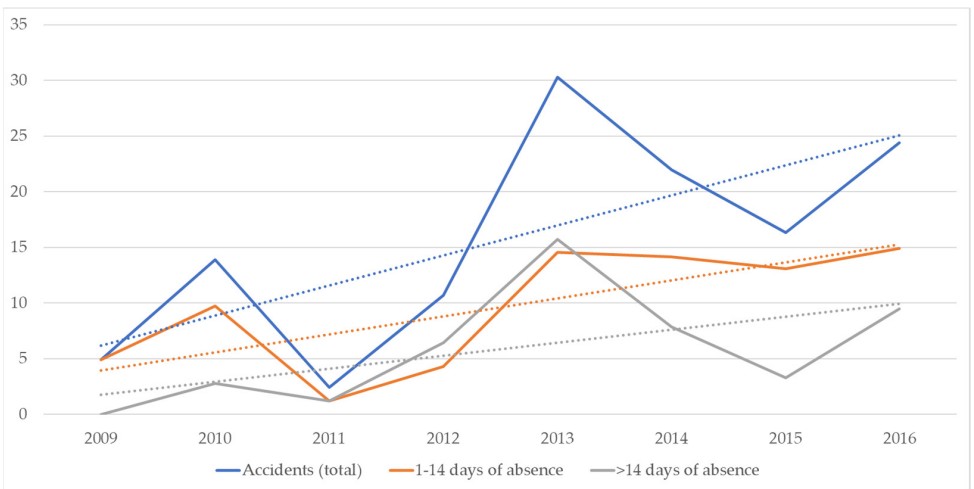

**Figure 7.** Incidence rates in construction of bridges and tunnels. Based on data from [17–24,27].

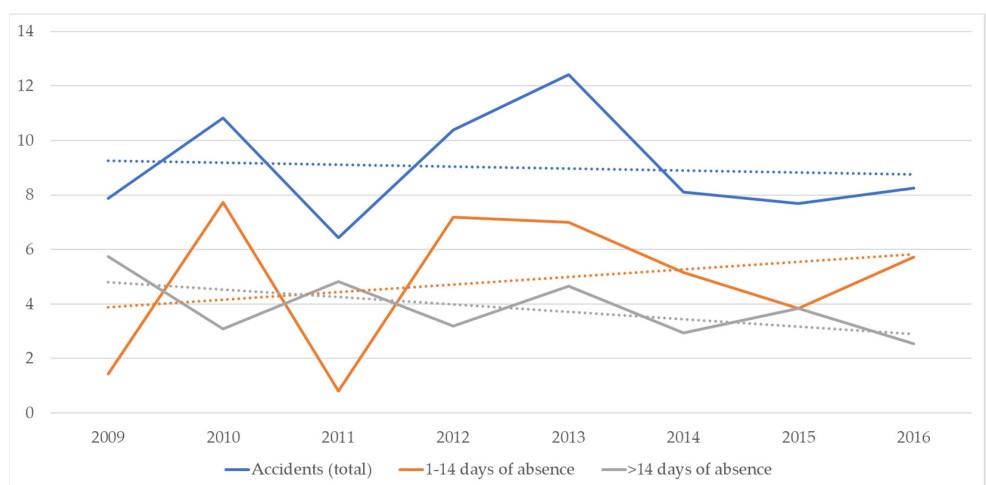

**Figure 8.** Incidence rates in construction of water projects/construction of other civil engineering projects n.e.c. Based on data from [17–24,27].

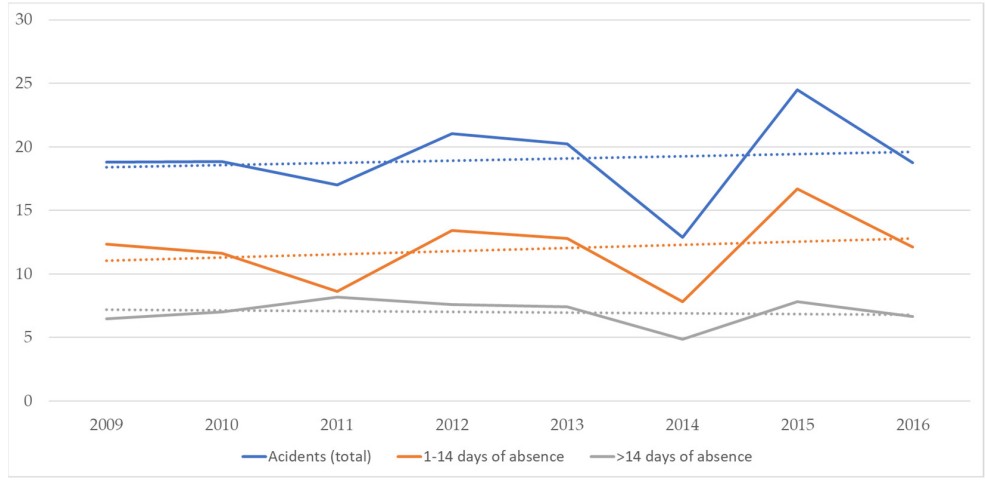

**Figure 9.** Incidence rates in construction of utility projects for fluids/construction of utility projects for electricity and telecommunications. Based on data from [17–24,27].

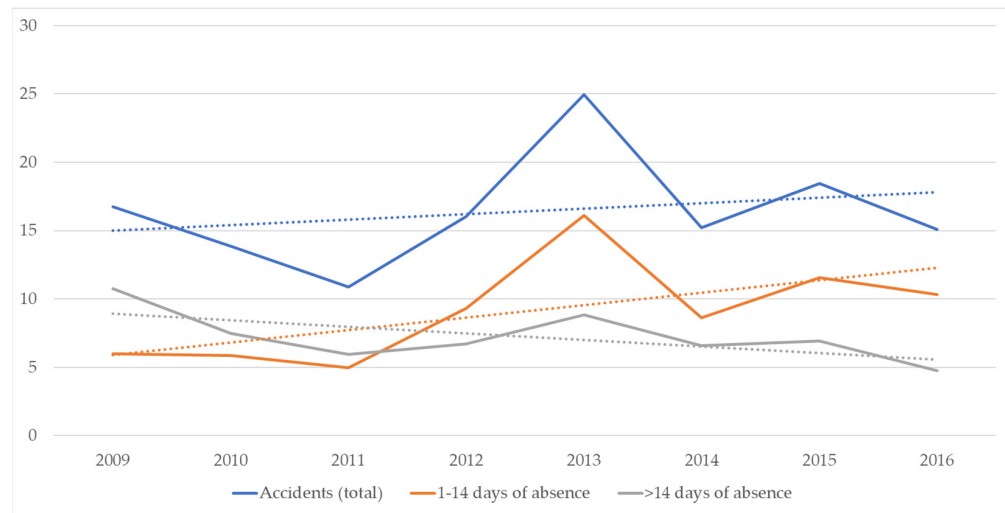

**Figure 10.** Incidence rates in demolition. Based on data from [17–24,27].

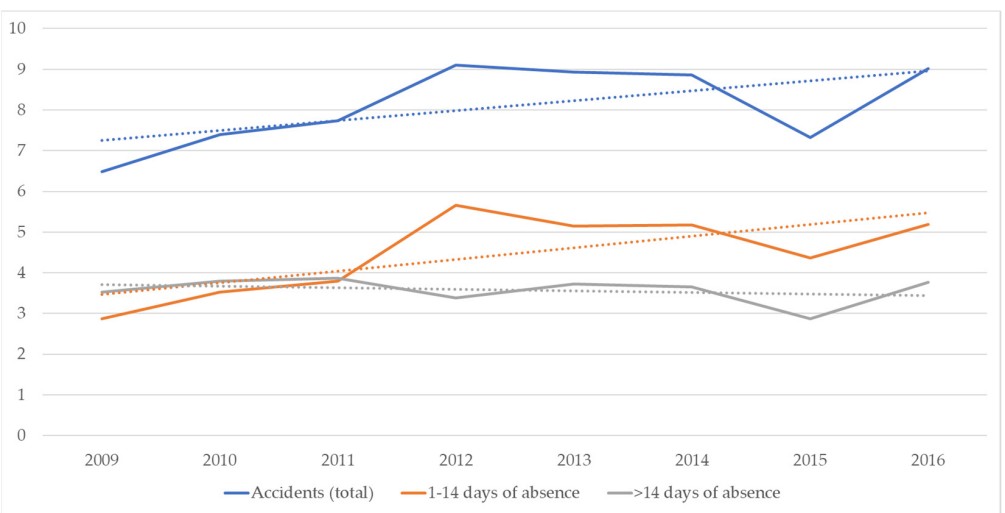

**Figure 11.** Incidence rates in site preparation/test drilling and boring. Based on data from [17–24,27].

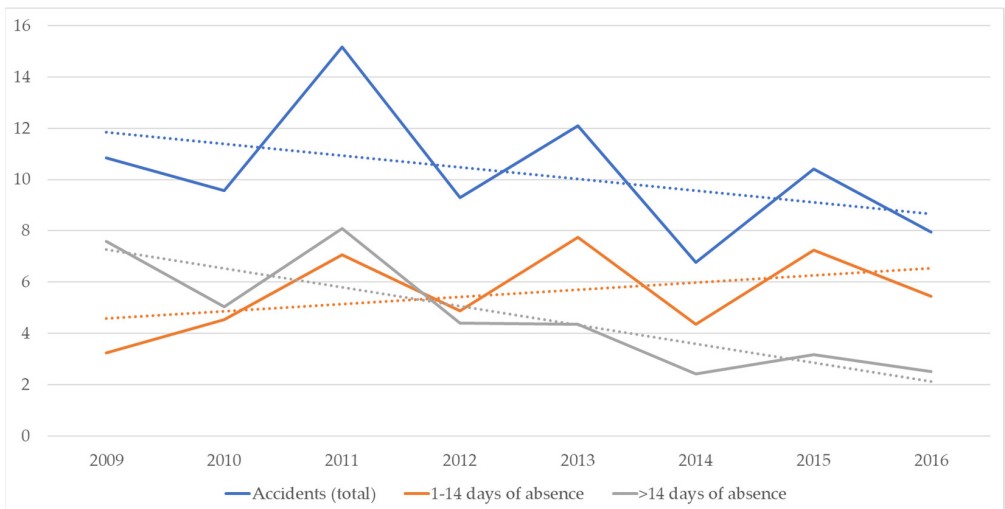

**Figure 12.** Incidence rates in plastering. Based on data from [17–24,27].

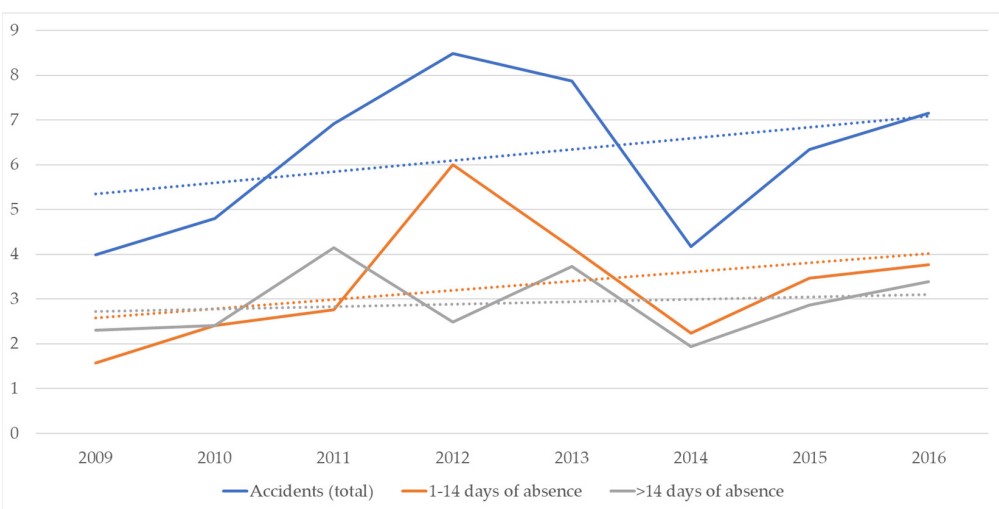

**Figure 13.** Incidence rates in floor and wall covering. Based on data from [17–24,27].

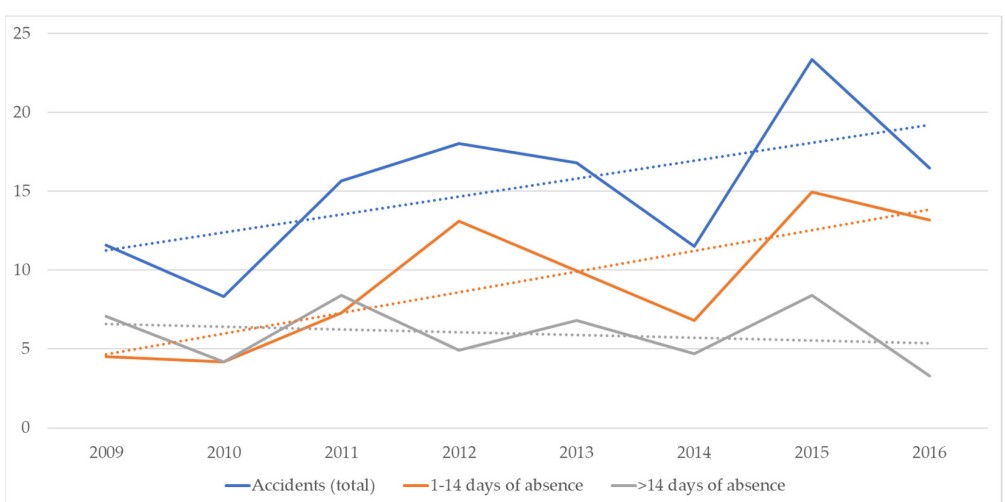

**Figure 14.** Incidence rates in erection of other roof covering and frames. Based on data from [17–24,27].

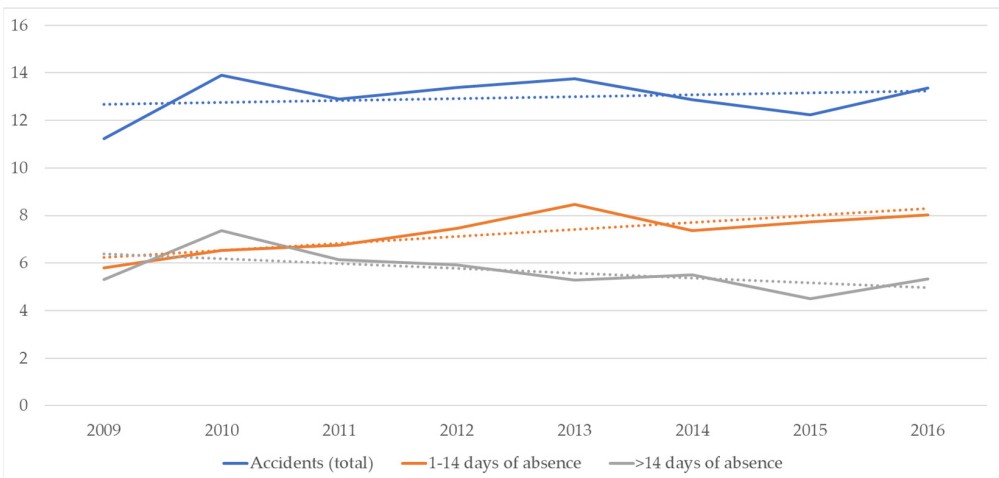

**Figure 15.** Incidence rates in various other specialised construction activities n.e.c. Based on data from [17–24,27].

*Trends in the Individual Subtrades between 2009 and 2016*

Taken together, an overall increase can be noted in the subtrades in question with increased incidence rates in nine out of the twelve subtrades. However, the size of the increase varies, which indicates that the development regarding accidents is slightly different in the various subtrades. The highest increase can be seen in construction of railways and underground railways and in construction of bridges and tunnels (Figures 6 and 7), wherein the incidence rates have gone from 4.01 to 22.61 and from 4.91 to 24.39, respectively. Site preparation/test drilling and boring (Figure 11), floor and wall covering (Figure 13), and erection of other roof covering and frames (Figure 14) also show an increase, although not as steep as the one mentioned before. A smaller increase is visible in development of building projects/construction of residential and non-residential buildings (Figure 4), various other specialised construction activities n.e.c. (Figure 15), construction of utility projects for fluids/construction of utility projects for electricity and telecommunications (Figure 9), as well as demolition (Figure 10). In contrast, in three of the subtrades, the incidence rate decreased, more specifically in plastering (Figure 12), construction of water projects/construction of other civil engineering projects n.e.c. (Figure 8), and construction of roads and motorways (Figure 5), wherein the latter stands out due to the trend of a decreasing incidence rate.

A further general trend in the individual subtrades is that short-term absence increased in relation to long-term absence. While long-term absence was higher in comparison to short-term absence in several subtrades in the beginning of the time series, it tends to be the opposite after 2011 or 2012. After 2011, short-term absence became more common than long-term absence in demolition (Figure 10), while long-term absence decreased from 10.77 in 2009 to 4.77 in 2016. Site preparation/test drilling and boring (Figure 11), floor and wall covering (Figure 13), erection of other roof covering and frames (Figure 14), and various other specialized construction activities n.e.c. (Figure 15) show similar patterns with an increase in short-term absence compared to long-term absence from 2011 and onwards. Within construction of water projects (Figure 8), it varied between long-term or short-term absence being the highest until 2012, when short-term absence increased rapidly. After that year, short-term absence was consistently higher. Plastering (Figure 12) also shows a similar pattern.

Some subtrades stand out as they do not fit the general trend. For example, within development of building projects (Figure 4) and construction of utility projects for fluids (Figure 9), long-term absence was constantly lower than short-term absence. That is also the case in construction of roads and motorways (Figure 5); from following each other quite closely, short-term absence began to increase in 2014 while long-term absence decreased. Within construction of railways and underground railways (Figure 6), short-term absence was generally higher compared to long-term absence, except for two points during 2012 and 2016.

## 4. Conclusions

The purpose of this article was to explore incidence rates in the subtrades that comprise construction and civil engineering between 2009 and 2016 and discuss these rates in relation to the introduction of new roles and responsibilities for health and safety management. The results show that the expected positive effects of this particular aspect of the EU directive 92/57/EEC were seemingly not achieved, and we observed not only a levelling out, but also a small increase in the rates in the years immediately following the introduction of the new regulations. Our study of the twelve different subtrades provides an insight into how different trades are related to different types of accidents in terms of length of absence from work, but they do not elucidate the basic question of why the accident rates did not decrease, but rather increased, as a consequence of the regulatory change.

One explanation may be that any or all potential positive effects of the new roles and responsibilities that were introduced in 2009 were counterbalanced by a change in reporting systems and financial indicatives, leading to more accidents coming into light.

When it comes to the system for reporting occupational injuries, it became possible from December 2011 and onwards to report injuries via the Internet, compared to the earlier system wherein the only option was to report on paper and send it in manually to the authorities. Another change took place in April of 2012 when the financial compensation for individuals suffering from occupational injuries changed so that they were compensated from day one [28]. Thus, it is possible that the tendency to report minor injuries increased due to the web-based forms and the possibility for the individual to receive financial compensation from the first day of absence. This receives support by the presented statistics which show that short-term absence increased in relation to long-term absence in almost all of the subtrades. In the beginning of the time series, long-term absence was higher in comparison to short-term absence in several subtrades, but after 2011/2012 it tended to be the opposite. This coincides with the implementation of the new system for reporting occupational accidents during those years.

Naturally, based on the data presented in this study, it is difficult to conclude whether the above-mentioned structural changes had an impact on the incidence rates in the specific subtrades that were studied. Indeed, as argued by Martinéz-Aires et al. [5], the sheer number of different health and safety initiatives and other developments in any given European country over the last decades makes is near impossible to tie accident trends to a specific regulation. However, the simultaneous rise in incidence rates and overarching changes to certain occupational health and safety practices in the Swedish construction industry deserve to be the focus of future studies. As argued by Al-Aubaidy et al. [29], it is vital to study mechanisms underpinning the underreporting of safety incidents and identify practices that may lead to more injuries coming into light in the construction industry. In that vein, a focus on changes to management structures and processes, including the digitalization of reporting systems and the introduction of financial incentives through insurances, is a fruitful avenue for further research.

For example, focus may be placed on expanding on the research conducted by Dellve et al. [30] and Almén and Larsson [6]. This could involve qualitative studies focusing specifically on the twelve subtrades and whether, e.g., injured workers have become more inclined to report accidents due to the improved reporting systems overall or, conversely, whether more accidents have occurred that at least in part may be connected to a lowered quality of health and safety management due to the implementation difficulties connected to the new roles and responsibilities. Case studies can also be designed to analyse specific practices that many of the subtrades share, such as scaffolding [31], i.e., take the complexity of these types of multi-employer worksites into account when it comes to accident reporting and day-to-day management. This could further the understanding of the impact of structural changes in the form of broader, standardised management practices [32] on incidence rates in the subtrades that comprise construction and civil engineering. Fundamentally, this provides an opportunity to analyse and discuss the structural changes that are nation-specific in relation to arguably the most significant change in health and safety policy in the European construction industry as a whole in recent decades, i.e., the EU directive 92/57/EEC.

Finally, although accidents and overall safety performance are reasonable starting points for such qualitative studies, the matter of occupational diseases and related trends also deserve further attention. Occupational diseases can be defined as "diseases which is the result of, or has gotten worse because of, work or working conditions" [33]. They are often caused by heavy workload or inappropriate work positions, unilateral work, exposure to dangerous substances, exposure to vibrations, and exposure to noise and/or mentally stressful work. For the individual, they can cause several health problems, such as back pain, eczema, allergies, mental illness, stomach ulcers, and heart problems [34]. Overall, occupational diseases affect the person as well as the companies and the social security systems. In Sweden, about five percent of the total workforce works within construction-related occupations, but this accounts for about 30 percent of all approved occupational diseases. Hence, employees working in the construction industry are identified as a group

that is more likely to suffer from occupational diseases compared to other industries [33]. Occupational disease trends thus become important to study as well, including their relation to changes to management structures and processes, imposed through supranational directives, i.e., how these have been perceived and adopted in practice with specific health issues in mind. In these cases, a focus on smaller companies may be especially important, given the need to improve the health and safety-related preventive measures that smaller companies undertake in general [35–37].

**Author Contributions:** Conceptualization, L.B., J.J., M.J., M.N. and M.S.; Methodology, L.B., M.N. and B.S.; Software, L.B.; Validation, L.B., J.J., M.J., M.N., B.S. and M.S.; Formal Analysis, L.B., J.J., M.J., M.S. and B.S.; Investigation, B.S.; Resources, B.S.; Data Curation, L.B. and B.S.; Writing—Original Draft Preparation, L.B., J.J., M.J., M.N. and M.S.; Writing—Review and Editing, M.N. and M.J.; Visualization, L.B. and M.N.; Supervision, J.J.; Project Administration, J.J. All authors have read and agreed to the published version of the manuscript.

**Funding:** This research received no external funding.

**Institutional Review Board Statement:** Not applicable.

**Informed Consent Statement:** Not applicable.

**Data Availability Statement:** All the statistics used for this article, specifically the number of accidents and the number of employees for the period of 2009 to 2016, are available online (https://ec.europa.eu/eurostat/databrowser/view/HSW_N2_01__custom_3507783/default/table?lang=en, accessed on 5 October 2022).

**Conflicts of Interest:** The authors declare no conflict of interest.

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
