# Peer review of "A Post-Analysis of the Introduction of the EU Directive 92/57/EEC in the Swedish Construction Industry"

_buildings, doi:10.3390/buildings12101765_

Round 1
Reviewer 1 Report
Please provide data source references for figures 1 and 2, are they from the same source mentioned in line 108, if so, please mention the source in the beginning, or are they from line 128? There is confusion about the data source; please ensure all figures have the proper data source.
This paper represents existing statical data (even then, there is confusion about the source), with many charts, the results are subjective, and there is a chance they only represent the authors’ opinions. It is strongly suggested, after the facts presented in the paper, the results should be supported by surveys, maybe interviews.
Reviewer 2 Report
1- The manuscript does not contain new information justifiable for publication and does not represent a good advance in knowledge.
2- In the introduction part, the existing work are described and referenced without a proper logic. A more targeted overview and summary according to the topic of this paper need to be conducted again.
The literature review seems is not be thoroughly done, so the paper is not very successful in elaborating the knowledge gap that the study is trying to fill. The flow of information on why this study is needed and how it builds on previous work and existing practice is unclear. Such a statement is very important for readers understanding, and should be presented clearly in Abstract, Introduction, and Conclusion. Specific to the literature review, the gap should be seen as the point of departure for the study. A revision is suggested to make the knowledge gap and the main contribution of the paper clear. Please discuss the limits of the literature and explain how you fill some of the existing gaps.
3- The methodology needs revisions and alone is not enough to analyze and develop the results and is not an innovation. Developing a hybrid model can provide better innovation. Reviewing and evaluating the selection and application of decision-making methods can be effective in providing a comprehensive hybrid model and provide better innovation. Also, the validation of the results requires more detailed descriptions and citations.
4- Conclusion section shall be improved indicating the findings in this article. In the current form, conclusion is very generic explanation. Try to be focused on what has been promised to deliver and what has been delivered. Conclusions should better reflect the paper contents and the results obtained in this article.
5- The results of the paper have a relatively practical process, but it is necessary for the authors to provide more explanations about the use of the results of different sections so that the audience can use it easily.
6- The language in the manuscript is weak and requires significant corrections to make it suitable for publication.
Reviewer 3 Report
The paper approaches a topical issue presenting an analysis of the effects of the EU directive 92/57/EEC on the work accidents rates in construction industry in Sweden.
The cited references are relevant and many of them are published in the last 5 years.
However, the paper needs to be improved because many aspects of the study are unclear.
General comments:
1) In Introduction section, more information about work accidents rates in construction industry at EU level should be presented, as well as the situation of Sweden in comparation with others EU members.
2) In order to justify the connection between the evolution of work accidents presented in figures 1 and 4 to 15, and EU directive 92/57/EEC, more information is needed. Thus, relevant information is missing, such as how the construction companies implement the directive, what is their level of compliance with directive’s provisions and occupational health and safety national legislation, training and awareness of workers, migrant workers or changes of the technologies. Such factors have an important influence on the work accidents rates, which is not related directly with the directive.
3) The EU directive 92/57/EEC is intended to prevent both work accidents and occupational diseases but the paper analyses only the work accidents. Thus, in order to achieve the objective proposed by the title of the paper, the evolution of occupational diseases in construction industry should be also taken into account.
4) Considering the missing information, the conclusion of the paper is not entirely sustained by the used methodology and obtained results.
Round 2
Reviewer 1 Report
Some trades show improvement over the years (Figures 5 and 12, for example). We need an explanation for why accidents in Road and Motorway construction decreased drastically. Is this related to EU laws, or is something else happening? The explanation in lines 227-230 is not justifying it.
There are a couple of abbreviations that the general users may not be familiar with (such as EU 15).
The article's title must change to reflect that this study only covers the case of Sweden, such as "in the Swedish Construction Industry."
The facts stated can not be generalized to other countries (Polland and Sweden, for example, could not have similar safety issues as their history for safety perceptions are different).
There is a need for the Conclusion section.
Reviewer 2 Report
I thank the authors for their efforts in improving the manuscript and applying the reviewers' comments.
Reviewer 3 Report
All my observations have been properly addressed. Some minor text editing corrections should be made, such as:
- text alignment to justify, according to the template;
- Row 65 - square brackets for the references ;
- Rows 95-98 - rows should be eliminated.
